# Fine-Tuning of GLI Activity through Arginine Methylation: Its Mechanisms and Function

**DOI:** 10.3390/cells9091973

**Published:** 2020-08-26

**Authors:** Yoshinori Abe, Nobuyuki Tanaka

**Affiliations:** Department of Molecular Oncology, Institute for Advanced Medical Sciences, Nippon Medical School, 1-1-5 Sendagi, Bunkyo-ku, Tokyo 113-8602, Japan; yoshiabe@nms.ac.jp

**Keywords:** Hedgehog signaling pathway, GLI family, protein arginine methyl transferase, signal transduction, stem cell

## Abstract

The glioma-associated oncogene (GLI) family consists of GLI1, GLI2, and GLI3 in mammals. This family has important roles in development and homeostasis. To achieve these roles, the GLI family has widespread outputs. GLI activity is therefore strictly regulated at multiple levels, including via post-translational modifications for context-dependent GLI target gene expression. The protein arginine methyl transferase (PRMT) family is also associated with embryogenesis, homeostasis, and cancer mainly via epigenetic modifications. In the PRMT family, PRMT1, PRMT5, and PRMT7 reportedly regulate GLI1 and GLI2 activity. PRMT1 methylates GLI1 to upregulate its activity and target gene expression. Cytoplasmic PRMT5 methylates GLI1 and promotes GLI1 protein stabilization. Conversely, nucleic PRMT5 interacts with MENIN to suppress growth arrest-specific protein 1 expression, which assists Hedgehog ligand binding to Patched, indirectly resulting in downregulated GLI1 activity. PRMT7-mediated GLI2 methylation upregulates its activity through the dissociation of GLI2 and Suppressor of Fused. Together, PRMT1, PRMT5, and PRMT7 regulate GLI activity at multiple revels. Furthermore, the GLI and PRMT families have strong links with various cancers through cancer stem cell maintenance. Therefore, PRMT-mediated regulation of GLI activity would have important roles in cancer stem cell maintenance.

## 1. Introduction

The Hedgehog (HH) signaling pathway is one of the most important pathways in tissue development and homeostasis [1,2]. However, constitutive HH signaling pathway activation has also been observed in many cancers, and promotes cancer cell proliferation, metastasis, and cancer stem cell (CSC) maintenance [3]. The glioma-associated oncogene (GLI) family was first identified as the downstream effector of the HH signaling pathway. The GLI family consists of three proteins—GLI1, GLI2, and GLI3—that all have five C_2_H_2_ Krüppel-like zinc finger motifs [4,5]. GLI1 only contains an activation domain at the C-terminus, whereas GLI2 and GLI3 have both a C-terminal activation domain and an N-terminal repressor domain. Among the GLI family of proteins, GLI2 is the major activator of HH signaling, whereas GLI3 is the major repressor. GLI1 most likely serves as a signal amplifier downstream of GLI2. Although the regulation of GLI proteins is well known to be regulated downstream of the HH signaling pathway [6], it has been reported that HH signaling pathway-independent regulation of GLI1 and GLI2 also exists [7]. The HH signaling pathway has a variety of outputs in embryogenesis, homeostasis, and cancers, and the GLI family is a key player that exhibits widespread outputs. Regulation of the GLI family therefore strictly occurs through multiple stages. A detailed understanding of the regulatory mechanisms of the GLI family is important for understanding the molecular mechanisms of embryogenesis, homeostasis, and oncogenesis.

The protein arginine methyl transferase (PRMT) family is also involved in embryogenesis and homeostasis through regulation of gene transcription, mRNA splicing, and stem cell function [8]. PRMTs expression is often elevated in cancer, and overexpressed PRMTs correlate with poor patient prognosis [9]. PRMTs are involved in cancer development through oncogenic signal transduction, epithelial-mesenchymal transition (EMT), and CSC maintenance [10]. Interestingly, the GLI family and PRMT family regulate normal and CSC function, which act as the origin of tissue and cancer development. Recent reports revealed PRMT1, PRMT5, and PRMT7 regulate GLI1 and GLI2 activity in normal and cancer cells. In this review, we discuss the roles of PRMT-mediated GLI regulation in relation to mainly CSC maintenance.

## 2. Regulation of GLI Transcriptional Activity

### 2.1. Regulation of GLI Proteins by HH-Dependent and -Independent Signaling Pathways

HH signaling is initiated by the HH ligands Sonic hedgehog (SHH), Indian hedgehog (IHH), and Desert hedgehog (DHH). The 12-pass transmembrane protein Patched (PTCH) is a receptor for HH ligands. In the absence of HH ligands, PTCH is localized to primary cilia and suppresses the HH signal transducer Smoothened (SMO), a seven-pass transmembrane protein [11]. Furthermore, GLI binds to Suppressor of Fused (SUFU), a suppressor of the HH signaling pathway [12,13] (Figure 1A). The SUFU–GLI complex is suppressed to accumulate in primary cilia and nuclei. Furthermore, GLI2 and GLI3 undergo proteolytic cleavage into repressor forms (GLI2R and GLI3R) [6]. After one of the three HH ligands binds to PTCH (Figure 1B), with assistance from the cell-surface proteins cell adhesion molecule-related/downregulated by oncogenes (CDO), brother of CDO (BOC), or growth-arrest-specific 1 (GAS1) [14], SMO accumulates in primary cilia and is fully activated. When the HH signaling pathway is activated, SUFU is dissociated from GLI2 and GLI3, and GLI2 and GLI3 accumulate in the nucleus and are fully activated (GLI2A and GLI3A). The activated GLI2 and GLI3 then induce GLI1 expression.

In addition to the canonical HH signaling pathway, tumor suppressors and several other signaling pathways are also involved in the regulation of GLI activity, independent of canonical HH signaling. In intact cells and cancer cells, the well-known tumor suppressor p53 represses GLI1 activity, and GLI1 also suppresses p53 activity [16,17]. Downregulation of p53 is mediated by the activation of mouse double minute 2 (MDM2) via AKT [16]. AKT is required for GLI1 activation [18,19], and an AKT repressor, phosphatase, and tensin homolog deleted from chromosome 10 (PTEN) also suppress GLI1 activity [19]. Various signaling pathways, including transforming growth factor-β (TGF-β) [20,21], receptor tyrosine kinases (RTKs), extracellular signal-regulated kinases 1 and 2 (ERK1/2) [22,23], and AKT [18,19], are involved in HH–SMO-independent GLI activation. These signaling pathways are all associated with cancers. Therefore, both an aberrant HH–SMO-independent GLI activation pathway and a loss of tumor suppressor function can boost tumor development (reviewed in [7]). Furthermore, integration of the HH and epidermal growth factor receptor (EGFR) signaling pathways regulates neural stem cell maintenance, homeostasis, and cancer development via HH–EGFR signaling, leading to specific GLI target gene expression [24,25,26] (described in the following section).

### 2.2. Modulation of GLI Activity

The GLI family has a wide variety of outputs. The various outputs from GLIs are strictly regulated in at least three ways (Figure 2). The first is the regulatory mechanism for GLI target gene expression, by the GLI state. The GLI activator forms (GLI1, GLI2 and GLI3 activator forms: GLI^A^) and GLI repressor forms (GLI2 and GLI3 repressor forms: GLI^R^) can bind the same sequences. However, GLI^A^ binding promotes GLI target gene expression, while GLI^R^ binding represses GLI target gene expression. The balance of GLI^A^ and GLI^R^ is mainly regulated by gradients in the concentrations of HH ligands. In vertebrate neural tube formation and limb formation, HH signaling regulates a time- and concentration-dependent induction of several cell populations, mediated by the balance of GLI^A^ and GLI^R^ amounts [27,28]. The balance of GLI^A^ and GLI^R^ is therefore necessary for context-dependent GLI target gene expression. The second way in which GLI outputs are regulated is through the integration of multiple signaling pathways. For example, GLI1 and c-JUN/activator protein 1 (AP-1) are activated downstream of the integration of the HH and EGFR signaling pathways, and induce HH–EGFR signaling pathway-specific GLI target gene expression. These target genes include cancer stemness genes (e.g., sex-determining region Y-box 9 (*SOX9*), fibroblast growth factor 19 (*FGF19*), and *SOX2*) [26], cytokines, and their receptor genes (e.g., interleukin-8 (*IL-8*) and IL-1 receptor, type II (*IL-1R2*)) [25,29]. The third way in which GLI outputs are regulated is through the fine-tuning of GLI activity by post-translational modifications [6,7] The N-terminus of GLI1 is phosphorylated by ERK1/2 [22,23], 40S ribosomal protein S6 kinase 1 (S6K1) [30] and atypical protein kinase C (aPKC) [31], and this GLI1 phosphorylation is associated with GLI1 activation. Thus, GLI1 activation signaling may be integrated on the N-terminus of GLI1. Protein kinase A (PKA) is a negative regulator of GLI. GLI2 and GLI3 have six direct sites of phosphorylation by PKA, behind the zinc finger motifs. The state of the six phosphorylation sites defines the GLI^A^ and GLI^R^ states [32]. The fully phosphorylated state drives GLI2 and GLI3 to have GLI^R^ functions. Partial dephosphorylation drives a weak GLI^A^ state, while GLI2 and GLI3 have GLI^A^ functions in the fully dephosphorylated state. In addition to phosphorylation-mediated GLI1 activation, the histone deacetylase 1 and 2 (HDAC1 and 2)-mediated GLI1 and GLI2 deacetylation of lysine residues (K518 in GLI1 and K757 in GLI2) also upregulates their transcriptional activities [33].

## 3. PRMTs

### 3.1. Overview of the PRMT Family of Proteins

PRMTs catalyze the transfer of a methyl group from S-adenosylmethionine (SAM) to the guanidino nitrogen atoms of arginine. This reaction results in the formation of methylarginine [34]. Currently, nine PRMTs have been identified, which can be classified into three categories depending on the type of arginine methylation that they perform [8] (Figure 3). Type I PRMTs (PRMT1, PRMT2, PRMT3, coactivator-associated arginine methyltransferase 1 (CARM1)/PRMT4, PRMT6, and PRMT8) catalyze the asymmetric dimethylation through monomethyl arginine of specific arginine residues. Type II PRMTs (PRMT5 and PRMT9) catalyze the symmetric dimethylation through monomethyl arginine of specific arginine residues. The type III PRMT (PRMT7) catalyzes only the monomethyl arginine of specific arginine residues. Arginine methylation by PRMTs is involved in gene transcription, pre-mRNA splicing, signal transduction, DNA damage response, and cell fate decisions [8,35,36].

### 3.2. Regulation of PRMT Activity by Post-Translational Modifications

PRMT activity is regulated by microRNA-mediated PRMT mRNAs stabilization and translation, ubiquitylation, and post-translational modifications [35]. Here, we summarize the regulation of PRMT activity by post-translational modifications. RhoA-activated kinase (RAK)- or liver kinase B1 (LKB1)-mediated phosphorylation of the N-terminus in PRMT5 upregulates its activity [37,38]. PRMT5 forms a complex with methylosome protein 50 (MEP50) to modulate its activity and substrate specificity [39,40]. Phosphorylation of MEP50 by cyclin-dependent kinase 4 (CDK4)-cyclin D1 also increases PRMT5 activity [41]. In contrast, JAK2V617F [42], a constitutive active mutant of Janus kinase 2 (JAK2) that is found in patients with myeloproliferative neoplasms, phosphorylates PRMT5 and downregulates its activity [43]. Furthermore, another member of the PRMT family, PRMT4, methylates PRMT5 and downregulates its activity [44]. It has also been reported that casein kinase 1 (CK1)-mediated PRMT1 phosphorylation regulates its specificity of chromatin binding [45], and that p38 mitogen-activated kinase (MAPK)-mediated PRMT4 phosphorylation regulates its localization [46].

### 3.3. Roles of PRMTs in Signaling Pathways

#### 3.3.1. Roles of PRMT1 in the TGF-β and Bone Morphogenetic Protein (BMP) Signaling Pathways

In the SMAD family of proteins, SMAD6 and SMAD7 have inhibitory effects on the BMP and TGF-β signaling pathways. PRMT1 is the primary type I PRMT, and accounts for approximately 90% of the global asymmetric dimethylated arginine generation [47]. PRMT1 is involved in the activation of these signaling pathways via the methylation of SMAD6 and SMAD7 [48,49,50]. Methylated SMAD6 is dissociated from BMP or TGF-β receptors, while methylated SMAD7 is degraded. SMAD1 and SMAD5 are then activated downstream of BMP receptors [48], or SMAD3 is activated downstream of TGF-β receptors. PRMT1-mediated SMAD3 activation induces EMT [49]. In addition, PRMT1-mediated SMAD6 methylation suppresses myeloid differentiation primary response 88 (MYD88) degradation [50]. Stabilized MYD88 is necessary for the activation of nuclear factor kappa-light-chain-enhancer of activated B cells (NF-κB), downstream of the Toll-like receptors (TLRs) signaling pathway.

#### 3.3.2. Roles of PRMT5 in the EGFR Signaling Pathway

PRMT5 methylates an arginine residue at 1175 in EGFR, which is contained in the intracellular domain of EGFR. This methylation induces autophosphorylation of a tyrosine residue at 1173 and the binding of Src homology region 2 domain-containing phosphatase 1 (SHP1) phosphatase. This interaction then attenuates the downstream rat sarcoma viral oncogene homolog GTPase (RAS)–ERK pathway [51]. PRMT5 also methylates the rapidly accelerated fibrosarcoma (RAF) family, B-RAF and C-RAF, which are RAS effectors [52]. Methylated B-RAF and C-RAF are destabilized and attenuate ERK1/2 activity. A higher activation state of ERKs induces differentiation, whereas the attenuation of ERK activity by PRMT5-mediated B-RAF and C-RAF methylation leads to cell proliferation. It is therefore assumed that PRMT5 in the EGFR and RAS–ERK pathways regulates ERK activity to select either cell proliferation or differentiation.

### 3.4. Roles of PRMTs in Genes Transcription via Histone Arginine Methylation

We introduce PRMT5-mediated gene transcription of an HH signaling activator in Section 4.3. Therefore, in the current section, we briefly describe the overview of PRMTs-mediated gene transcription through arginine methylation. Detailed mechanisms of PRMTs-mediated gene transcript are reviewed in [8,9].

PRMT1 and PRMT4 are known as transcriptional activators. PRMT1 asymmetrically methylates an arginine residue at 3 in histone H4 (H4R3me2a). Then, histone acetyltransferase p300/CREB-binding protein (CBP) is recruited to H4R3me2a and potentiates acetylation of lysine residues at 9 and 14 in histone H3, which facilitate the binding of transcription factors [53]. PRMT4 is a transcriptional coactivator that preferentially methylates arginine residues at 17 (H3R17me2a), 26 (H3R26me2a), and 42 (H3R42me2a) in histone H3. PRMT4-mediated histone modifications of H3R17me2a, H3R26me2a, and H3R42me2a are associated with reduced binding of the nucleosome remodeling and deacetylase complex [54].

PRMT5 also epigenetically regulates gene expression but can either drive or repress transcription according to which residues are methylated in the histone. PRMT5 symmetrically methylates an arginine residue at 3 in histone H4 (H4R3me2s). H4R3me2s serves as a direct binding target for the DNA (cytosine-5)-methyltransferase 3A (DNMT3A), and DNMT3A methylates neighboring DNA to repress the associated gene expression [55]. By contrast, PRMT5 promotes gene expression by methylation of an arginine residue at 2 in histone H3 (H3R2me2s). H3R2me2s is recognized by the WD40 domain of WD repeat 5 (WDR5), enabling the recruitment of the mixed-lineage leukemia (MLL) complex, trimethylation of a lysine residue at 4 in histone H3 [56].

## 4. PRMT-Mediated Regulation of GLI Activity

We introduced the role of PRMTs in signaling pathways in the previous section. In each of these pathways, a single PRMT regulates their activity. However, in the HH signaling pathway, multiple PRMTs (PRMT1, PRMT5, and PRMT7) regulate GLI activity through arginine methylation. Here, we introduce PRMT1-, PRMT5-, and PRMT7-mediated GLI regulation machinery (Figure 4).

### 4.1. Upregulation of GLI1 Transcriptional Activity by PRMT1-Mediated GLI1 Methylation

Pancreatic ductal adenocarcinoma (PDAC) has one of the worst prognoses of all cancers, and is known to reactivate embryonic signaling pathways. The HH signaling pathway has important roles in PDAC [57]. GLI1 is also regulated downstream of K-RAS and TGF-β but not the canonical HH signaling pathway, and is involved in PDAC cell survival and transformation [58]. Wan et al. [59] demonstrated that GL1 is methylated by PRMT1 in PDAC. PRMT1 was identified as a component of the GLI1 complex from MIA-PaCa2 PDAC cell lines. The authors reported that GLI1 directly interacts with PRMT1, but PRMT1 does not interact with GLI2 and GLI3, suggesting that PRMT1 exclusively regulates GLI1. In addition, immunohistochemical analysis revealed that the expression levels of PRMT1 and GLI1 are correlated in PDAC lesions.

PRMT1 upregulates GLI1 transcriptional activity via the methylation of an arginine residue at 597 in GLI1. This strongly induces the expression of some GLI1 target genes, such as insulin growth factor-binding protein 6 (*IGFBP6*) and B-cell lymphoma 2 (*BCL2*), that are involved in PDAC cell survival [60]. However, the expression of some GLI1 target genes, such as *PTCH1* and *CCND1* [5], is not altered by PRMT1-mediated GLI1 methylation. Interestingly, PRMT1-mediated methylation of GLI1 is independent of both the canonical HH signaling pathway and the SMO-independent GLI1 activation pathways, which are active in PDAC.

GLI1 is associated with tumor formation in PDAC, and GLI1 knockout in PDAC cells inhibits cell proliferation and prevents tumor formation. GLI1 expression in GLI1-knockout PDAC cells leads to the recovery of tumor formation ability. However, the expression of a GLI1 mutant, with the arginine residue at 597 substituted by lysine (GLI1-R597K mutant), does not lead to the recovery of tumor-forming abilities in GLI1-knockout PDAC cells. Therefore, PRMT1-mediated GLI1 methylation is necessary for the development of PDAC.

These results suggest that PRMT1-mediated GLI1 methylation regulates GLI1 activity and specific GLI1 target gene expression (*IGFBP6* and *BCL2*) to support PDAC cell survival and transformation.

### 4.2. GLI1 Protein Stabilization by Cytosolic PRMT5-Mediated GLI1 Methylation

GLI1 activity is mainly regulated downstream of SMO. However, the activation mechanisms of GLI1 in HH signaling after its dissociation from SUFU are not fully understood. MEP50 is also known as androgen receptor cofactor p44/WD repeat domain 77 (WDR77) [39,40], and acts as a cofactor of PRMT5, as a novel GLI1-binding partner in the cytoplasm [61]. GLI1 interacts with the PRMT5–MEP50 complex through WD repeats in MEP50, and the interaction between GLI1 and MEP50 is increased by HH signaling pathway activation. However, MEP50 does not interact with GLI2 and GLI3. The PRMT5–MEP50 complex methylates arginine residues at 515, 915, 940, 990, 1018, 1068, and 1091 in GLI1. However, the arginine residue at 597 in GLI1, which is methylated by PRMT1 [59], is not methylated by PRMT5. PRMT5-mediated arginine methylation at 990 and 1018 in GLI1 is necessary for GLI1 protein stabilization. GLI1 protein levels are regulated by the adaptor protein Numb, which recruits GLI1 to the E3 ubiquitin ligase ITCH [62,63]. GLI1 mutants, with the arginine residues at 990 and 1018 substituted with lysine, have enhanced interaction with the ITCH–Numb E3 ligase complex, resulting in GLI1 ubiquitination. Furthermore, in C3H10T1/2 cells, which are derived from mouse embryonic fibroblasts, *Prmt5* and *Mep50* mRNA levels are enhanced by HH signaling pathway activation, suggesting that PRMT5 and MEP50 are positive feedback regulators in HH signaling.

PRMT5-mediated GLI1 stabilization machinery has also been observed in the previously identified SHH-expressing gastric cancer cell line AGS [64] and the small-cell lung cancer (SCLC) cell line H146 [65]. siRNA-mediated PRMT5 or MEP50 knockdown inhibits anchorage-independent cancer cell proliferation. Furthermore, PRMT5 suppression sensitizes cells to the SMO inhibitor cyclopamine [66]. From a database analysis, PRMT5, MEP50, and GLI1 target genes are all upregulated in known HH signaling pathway-activated cancers, including SCLC [65], gastric cancer [64], skin basal carcinoma [67], and breast cancer [68]. Together, PRMT5-mediated GLI1 methylation activates GLI1 via GLI1 protein stabilization. Furthermore, PRMT5 and MEP50 expression is enhanced by HH signal activation, and they act as positive feedback regulators in the HH signaling pathway. It therefore appears that both aberrant HH signal activation and PRMT5 and MEP50 overexpression are associated with tumorigenesis and cancer development.

### 4.3. Indirect Suppression of GLI1 Activity by PRMT5–Menin Complex-Mediated Epigenetic Modification in Multiple Endocrine Neoplasia Type 1 (MEN1) Tumor Syndrome

As described in the previous section, cytosolic PRMT5 assists with GLI1 activation. In contrast, nucleic PRMT5 interacts with Menin and indirectly suppresses HH signaling in MEN1 tumor syndromes, such as pancreatic islet tumors [69]. The nucleic protein Menin is encoded by the *MEN1* gene, and suppresses cell cycle suppressor, such as p27Kip1 and p18INK4c [70,71,72]. MEN1 is also associated with tumor formation in mixed lineage leukemia as well as pancreatic islet tumors [73].

The HH signaling pathway begins with the binding of HH ligands to PTCH. In addition to PTCH, other HH ligands bind cell-surface proteins, such as CDO, BOC, or GAS1, to act as coreceptors to assist with HH signal reception. PRMT5 has been identified as a Menin-mediated transcriptional repressive partner. PRMT5 interacts with Menin and is required for the suppression of *GAS1* gene expression. PRMT5 is associated with H4R3me2s, and this modification acts to suppress gene expression [55]. Menin recruits PRMT5 to the *GAS1* promoter, and PRMT5 increases H4R3me2s at this gene. Interestingly, a mutant form of Menin that is found in mixed lineage leukemia reduces its interaction with PRMT5, resulting in a reduction of H4R3me2s and a failure to suppress *GAS1* gene expression. Moreover, MEN1 knockout in pancreatic islet cells enhances GAS1 and GLI1 gene expression.

### 4.4. PRMT7 Upregulates GLI2 Activity via Suppression of SUFU binding

SUFU is a major negative regulator of the HH signaling pathway. It interacts with GLI2 and prevents its nuclear localization, thereby inhibiting its transcriptional activity; a lack of SUFU leads to constitutive HH signaling pathway activation [12,13]. There is also PRMT7-mediated regulation of the interactions between GLI2 and SUFU [74], resulting in GLI2 activation. Because both the HH signaling pathway and PRMT7 suppress cellular senescence [75,76], the role of PRMT7 in the HH signaling pathway has been examined. PRMT7 is involved in the induction of GLI1 expression via HH signaling pathway activation. GLI1 expression is also known to be enhanced by GLI2. Transient PRMT7 expression enhances GLI2 activity, suggesting that PRMT7 regulates GLI2 downstream of the HH signaling pathway. Furthermore, activated GLI2 inhibits p16INK4a expression, resulting in the suppression of cellular senescence. PRMT7 also regulates GLI2 localization to primary cilia as well as nuclei in response to HH signaling pathway activation. The regulation of GLI2 localization is important for its activation. An analysis of the PRMT7-mediated regulatory mechanisms of GLI2 localization revealed that PRMT7 interacts with GLI2 and methylates arginine residues at 225 and 227 in GLI2. These arginine residues are proximal to the binding sites of SUFU [77], which suppresses GLI2 activity through the inhibition of GLI2 localization to primary cilia and nuclei. PRMT7-mediated GLI2 methylation attenuates the interaction between GLI2 and SUFU. Taken together, these results suggest that PRMT7 mediates GLI2 activation through the regulation of its localization downstream of the HH signaling pathway. They also suggest that PRMT7 is involved in the inhibition of cellular senescence. The inhibition of cellular senescence dismisses a safeguard against tumorigenesis. Therefore, the PRMT7-mediated inhibition of cellular senescence may be a novel mechanism of tumorigenesis and cancer formation.

## 5. Consideration of the Roles of PRMT-Mediated GLI Regulation in the Development and Cancers, with a Focus on Stem Cell Maintenance

The PRMT and GLI families are necessary for embryonic, tissue, and CSC maintenance. As mentioned in the previous section, PRMTs are involved in the regulation of GLI1 and GLI2 activity. Therefore, in the current section, we consider the roles of some PRMTs (PRMT1, PRMT5, and PRMT7) and the GLI family in normal and CSCs.

The PRMT family maintains stemness and differentiation through histone modifications and signal transduction. PRMT4 epigenetically upregulates key stemness genes (*NANOG*, octamer-binding transcription factor 4 (*OCT4*), *SOX2*) through histone H3 methylation (H3R17me2a and H3R26me2a) at their promotor [78]. PRMT5 is associated with upregulation of *OCT4* and *NANOG* by methylation of cytoplasmic histone H2A for maintaining pluripotency of embryonic stem cells (ESCs) [79]. PRMT5 epigenetically upregulates Forkhead box protein P1 (FOXP1) expression through histone H3 methylation (H3R2me2s) at this gene promotor for breast cancer stem cells’ maintenance [80]. PRMT7 epigenetically suppresses micro-RNA (miRNA) expression through histone H4 methylation at the *miR24-2* and *miR-221* promotor region, allowing the transcription of *OCT4*, *NANOG*, Krüppel-like factor 4 (*KLF4*), and *C-MYC* in ESCs [81,82]. PRMT1 is also required for the pluripotency of progenitor cells [45]. The HH signaling pathway is involved in the maintenance of stem or progenitor cells in many adult tissues, including in the epithelia of many internal organs and the brain [2]. For example, GLI1 is involved in neural stem cell maintenance via the upregulation of stemness genes, such as *NANOG*, *KLF4*, B cell-specific Moloney murine leukemia virus integration site 1 (*BMI1*), and *SOX2* [17].

Arginine methylation also plays a role in the reprograming of somatic cells into induced pluripotent stem cells (iPSCs). PRMT5 cooperates with Yamanaka factors (OCT4, SOX2, KLF4, C-MYC) [83] and acts as a reprograming factor [84]. PRMT5 enhances the generation of iPSCs by downregulating p53, p21, and caspase 3 signaling [85]. In addition, PRMT5 and PRMT7 are capable of replacing SOX2 in the generation of iPSCs [84,86]. GLI1 is also involved in iPSC generation by upregulating the expression of a GLI1 target gene, *BMI1* [87]. BMI1 induces three of the four Yamanaka factors (SOX2, KLF4, and N-MYC). GLI1 activation may therefore improve the efficiency of iPSC generation.

The GLI and PRMT families are also involved in CSC maintenance by upregulating the expression of each target gene. In the acute myeloblastic leukemia with maturation (M2) subtype of acute myeloid leukemia (AML), the in-flame fusion protein AML1-ETO (AE) isoform AE9a, a transcription factor, interacts with PRMT1 and recruits it to the AE9a promoter. PRMT1 then induces histone modification proximal to the AE9a promoter. Finally, PRMT1 is involved in the induction of AE9a target stemness genes in AML [88]. PRMT5 is overexpressed in chronic myeloid leukemia stem cells and breast cancer stem cells, and is required for CSC proliferation and self-renewal [80,89]. PRMT5 appears to epigenetically regulate CSC gene expression via a common mechanism. However, the specific genes that are epigenetically regulated appear to be tumor-type specific [80,89]. EMT is a process linked to the acquisition of stem cell-like properties [90]. PRMT1 activates SMAD3 by preventing SMAD7 activity downstream of the TGF-β signaling pathway, and promotes EMT [49]. PRMT5 also regulates EMT through histone modification [91,92]. The GLI family maintains CSCs in various cancers (e.g., glioblastoma, non-small cell lung cancer, chronic myeloid leukemia, and colon cancer) via the regulation of various target genes that are associated with CSC maintenance, including *SOX2*, *NANOG*, and *BMI1* [93,94,95]. As well as the PRMT family, GLI1 and GLI2 are also associated with EMT through their target gene expression [96,97,98]. Cancer cells are genetically and epigenetically plastic, suggesting that proliferative cancer cells can be reprogramed into other lineages, such as CSCs. A common target gene of PRMTs and GLI, *SOX2*, is a one of the factors that reprogram differentiated cancer cells into CSCs [99]. Reprogramed CSCs have similar abilities to initiate tumor growth and metastasis as primary CSCs, and possess similar gene profiles [99,100]. Therefore, PRMTs and GLI may be involved in the reprograming of cancer cells.

From the accumulated evidence, we here consider the roles of PRMT-mediated GLI1 or GLI2 activity regulation (Figure 5). Among the stemness genes, *SOX2* and *NANOG* are common target genes of PRMTs and GLI. However, although PRMTs and GLI seem to enhance the expression of these genes using different machinery, PRMT-mediated GLI regulation may be involved in regulating these expressions in a context-dependent manner. The expression of both PRMT and GLI target genes is dependent on the context, such as tissue or tumor specificity. The selection of target gene expression is modulated depending on upstream signaling and the integration of multiple signaling pathways. It is therefore possible that PRMTs and GLI cooperate with one another and are associated with the regulation of context-dependent gene expression. For example, PRMT5-mediated GLI1 stabilization seems to be involved in the balance of GLI^A^ and GLI^R^, depending on the HH signaling strength; PRMT1-mediated regulation of GLI1 activity is involved in the fine-tuning of GLI activity and the expression of specific target genes, like the regulation of specific GLI target gene expression through the integration of HH and EGFR signaling; and PRMT7-mediated GLI2 activation seems to also fine-tune GLI2 activity to enhance context-dependent GLI2 target gene expression. Furthermore, PRMT-mediated chromatin modification may regulate GLI target gene expression. Taken together, PRMTs regulate GLI activity on multiple levels. This novel GLI regulatory machinery likely has an important role in tissue development and oncogenesis via stem cell maintenance, including the reprograming of differentiated cells.

## 6. Outlook for PRMT-Mediated GLI1 Regulatory Mechanisms

PRMT1, PRMT5, and PRMT7 are newly identified regulators of GLI. They regulate GLI1 stabilization and GLI1 and GLI2 transcriptional activity in both intact cells and cancer cells. As mentioned in Section 4.1, PRMT1-mediated methylation of GLI1 is independent of both the canonical HH signaling pathway and the non-canonical GLI1 activation pathways, which are active in PDAC. Therefore, upstream of the PRMT1–GLI1 axis remains to be elucidated. From the analysis of upstream regulators of the PRMT1–GLI1 axis, a novel regulatory mechanism may be discovered in PDAC formation. PRMT5 methylates arginine residues at 990 and 1018 in GLI1 to stabilize GLI1 proteins. In addition, PRMT5 methylates six other arginine residues in GLI1. Because the roles of these additional methylations have not yet been defined, it would be interesting to understand the roles of other arginine residues that are methylated by PRMT5. Recently, it was reported that PRMT4 regulates PRMT5 activity [44]. Therefore, PRMT4 may be also involved in the regulation of GLI1 activity. PRMT5 acts as a regulator in growth factor signaling pathways. Therefore, PRMT5 could integrate GLI activation pathways. GLI3 is believed to mainly act as a suppressor of the HH signaling pathway [6]. Whether GLI3 activity is regulated by PRMTs remains to be elucidated. Since GLI activity is regulated by multiple PRMTs, PRMTs would be important regulators for GLI activity. Further research focusing on the interactions between PRMTs and GLI will provide insight into the regulatory mechanisms of GLI that enable a wide variety of outputs.

Aberrant HH signaling pathway activation is associated with a variety of cancers [3]. Several SMO inhibitors have therefore been developed, and two of them (Vismodegib/Eribedge^®^ [101] and Erismodegib/Sonidegib/Odomzo^®^ [102]) have already been approved by the US Food and Drug Administration for treating basal cell carcinoma. In addition, clinical trials of SMO inhibitors are ongoing in various HH signaling pathway-activating cancers. However, treatment with SMO inhibitors has led to the emergence of SMO inhibitor-resistant mutations [103,104,105]. Furthermore, SMO-independent GLI activation has been reported in some cancers, including non-small cell lung cancer and PDAC [7,58,106]. GLI inhibitors have therefore also been developed, such as GANT-58, GANT-61 [107], and arsenic trioxide [108,109]. Clinical trials using arsenic trioxide are about to begin in patients with advanced basal cell carcinoma (NCT02699723). PRMTs are also associated with various cancers [110], including AML [88,111,112], B-cell lymphoma [113], lung cancer [114,115], and breast cancer [80]. Currently, many PRMT inhibitors have been developed [110,116], and some of them have entered or are preparing to enter clinical trials as cancer therapies [35]. Thus, suppression of PRMT-mediated GLI regulatory machinery using these inhibitors may reduce CSC populations, and may also induce the reprograming of CSCs or cancer cells into benign cells.

## Figures and Tables

**Figure 1 cells-09-01973-f001:**
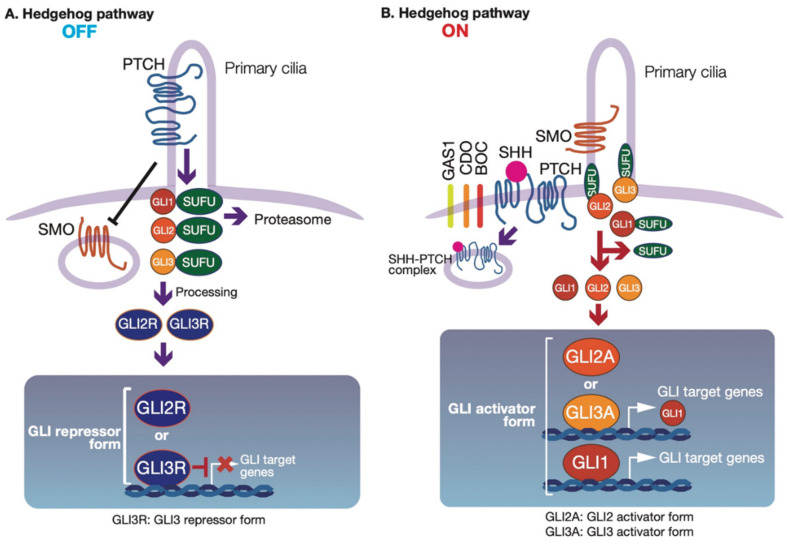
Hedgehog (HH) signaling pathway. (**A**) In the absence of HH ligands (the OFF state), Patched (PTCH) inhibits Smoothened (SMO) from primary cilia, and Suppressor of Fused (SUFU) binds to glioma-associated oncogenes (GLIs) and inhibits their localization to primary cilia and nuclei. SUFU binding of GLI1 and GLI2 is degraded, leading to the GLI2 or GLI3 repressor forms, followed by N-terminus processing of GLI3 and a part of GLI2, suppressing GLI target gene expression. (**B**) In the presence of HH ligands (the ON state), one of the HH ligands binds to PTCH and prevents PTCH from inhibiting SMO. Next, SMO translocates to primary cilia to be fully activated. SMO then mainly activates GLI2, and the activated GLI2 induces GLI1 expression. GLI1 is also activated downstream of SMO. Activated GLI2 (GLI2A) and GLI1 further upregulate the expression of various GLI target genes. Only core components of the HH signaling pathway are shown in this figure, which is modified from our previous work [15].

**Figure 2 cells-09-01973-f002:**
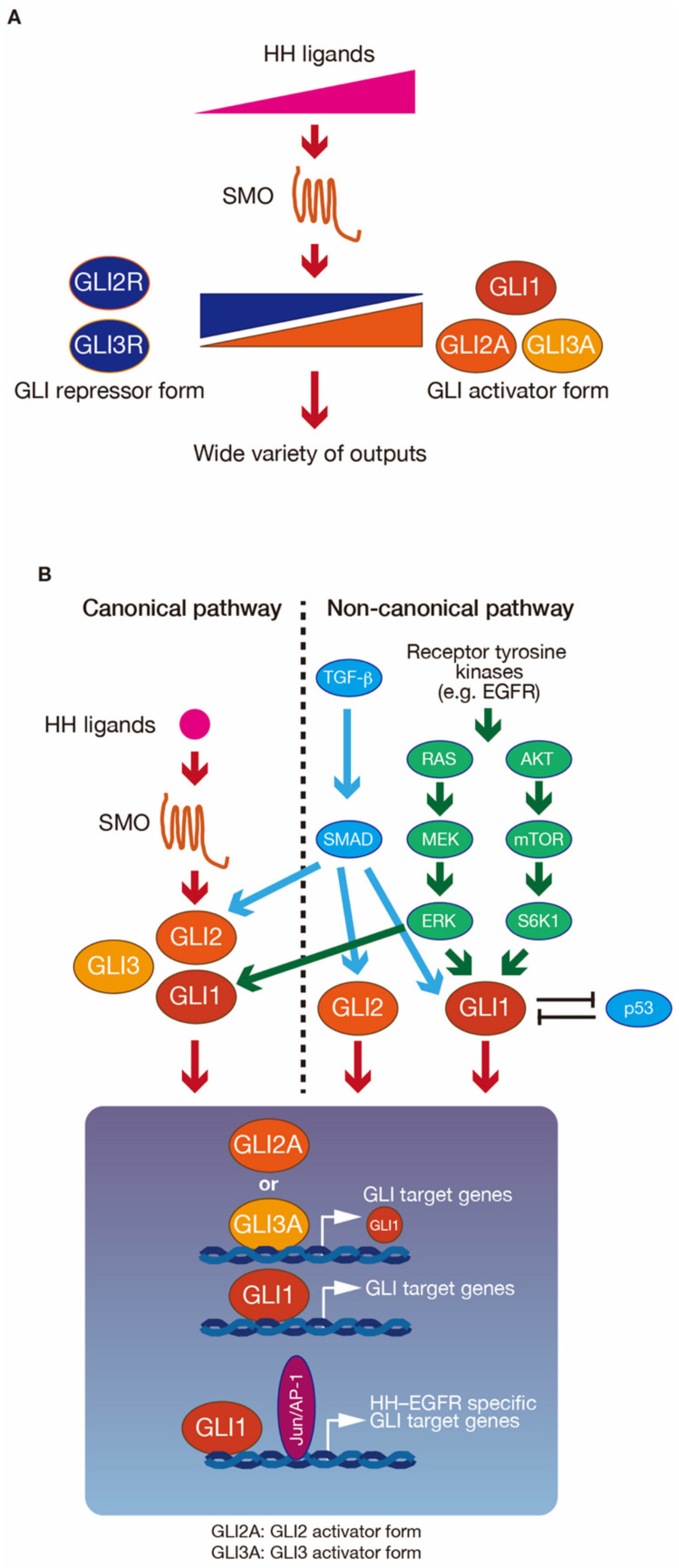
Modulation of glioma-associated oncogene (GLI) activity for a wide range of outputs. (**A**) Gradient Hedgehog (HH) ligand concentrations modulate the amount of GLI in the activator form (GLI^A^) and the repressor form (GLI^R^). Different ratios of GLI^A^ to GLI^R^ are required for various outputs. (**B**) GLI activity is regulated through both canonical and non-canonical signaling, as well as through the integration of multiple signaling.

**Figure 3 cells-09-01973-f003:**
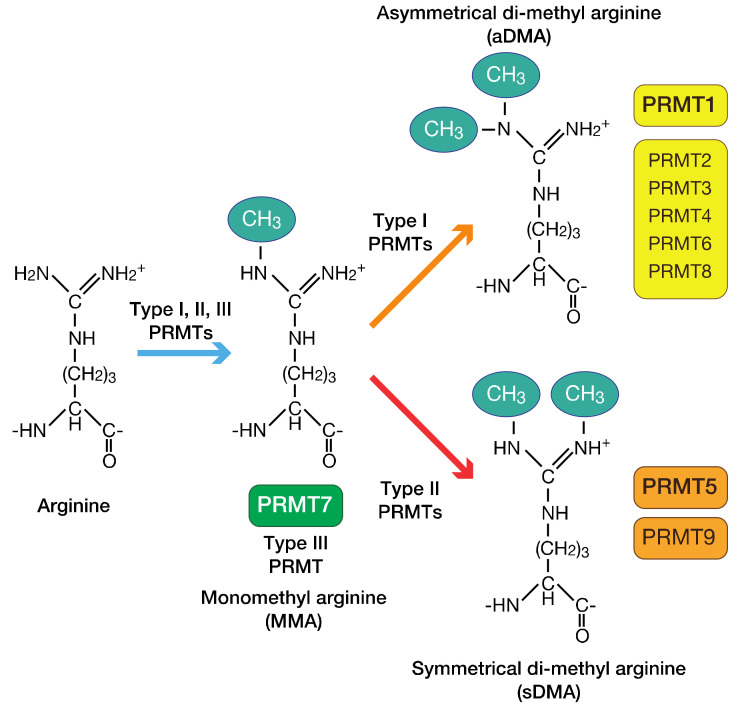
Protein arginine methylation. All of the protein arginine methyl transferase (PRMT) family of proteins catalyze monomethylated arginine. Type I PRMTs then catalyze asymmetrical methylated arginine, whereas type II PRMTs catalyze symmetrical methylated arginine. The type III PRMT catalyzes only monomethyl arginine.

**Figure 4 cells-09-01973-f004:**
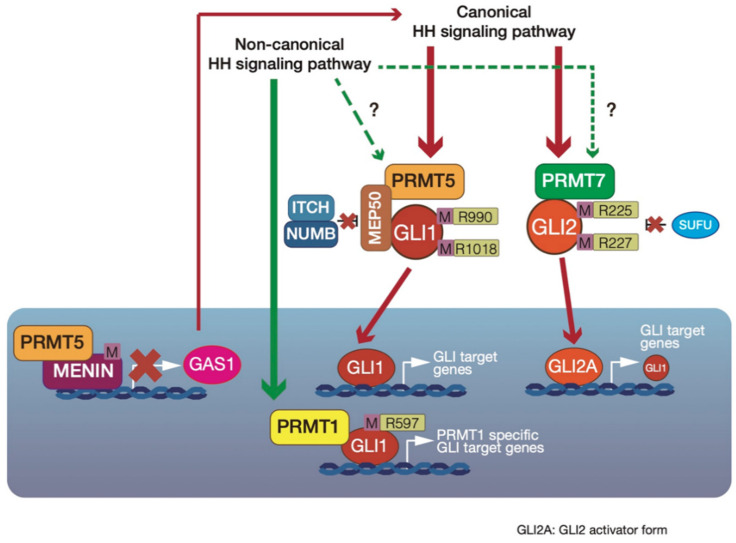
Protein arginine methyl transferase (PRMT)-mediated modulation of glioma-associated oncogene (GLI) 1 or GLI2 activity. PRMT1 methylates an arginine residue at 597 in GLI1. This PRMT1-mediated GLI1 methylation assists with the upregulation of GLI1 activity. Furthermore, methylated GLI1 enhances the expression of some specific GLI1 target genes (*IGFBP6* and *BCL2*). Cytoplasmic PRMT5 methylates arginine residues at 990 and 1018 in GLI1. This PRMT5-mediated GLI1 methylation inhibits the interactions of GLI1 and ITCH–Numb E3 ligase, and stabilizes the GLI1 protein. In contrast, nucleic PRMT5 interacts with Menin and represses growth arrest-specific protein (GAS) 1, and indirectly downregulates the HH signaling pathway. PRMT7 methylates arginine residues at 225 and 227 in GLI2. This PRMT7-mediated GLI2 methylation inhibits the binding of Suppressor of Fused (SUFU) to GLI2, resulting in GLI2 activation.

**Figure 5 cells-09-01973-f005:**
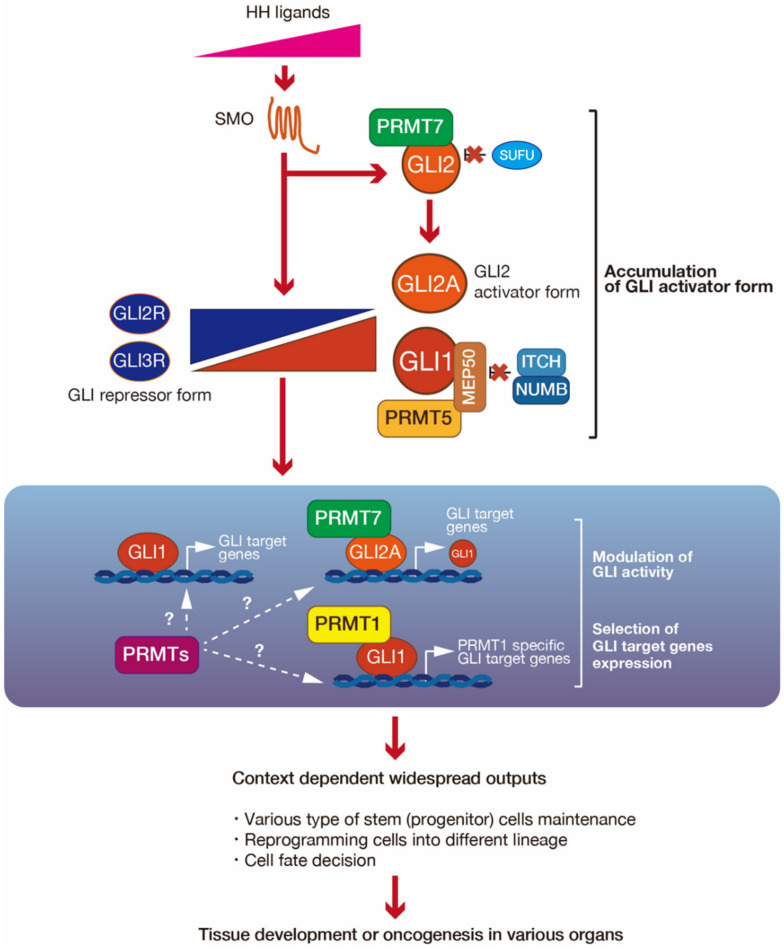
Possible roles of glioma-associated oncogene (GLI) 1 and GLI2 regulation by protein arginine methyl transferase (PRMT) 1, PRMT5, and PRMT7. PRMT1, PRMT5, and PRMT7 regulate GLI1 and GLI2 activity on multiple levels. PRMT1 modulates GLI1 activity and context-dependent GLI1 target gene expression. PRMT5 and PRMT7-mediated GLI regulation would modulate the ratio of GLI^A^ and GLI^R^. PRMTs may also regulate GLI target gene expression via histone modifications. This regulatory machinery may regulate the widespread context-dependent outputs of GLI.

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
