# Peer review of "Fine-Tuning of GLI Activity through Arginine Methylation: Its Mechanisms and Function"

_cells, 2020, doi:10.3390/cells9091973_

Round 1
Reviewer 1 Report
The review of Abe and Tanaka is well written and interesting since the modulation of GLI proteins by PRMTs is novel. The focus on cancer is also interesting since a lot of work is in place to develop inhibitors of PRMTs as anti-cancer agents.
However, the review is mainly focused on cancer and thus maybe in cancer stem cells rather than in stem cells. This should be better explained in the Abstract and Introduction.
- In Figure 1 the repressor forms have been enclosed and in the text of 2.1. However, the explanation on how these repressor forms are formed is delayed on 2.2. The authors should anticipate in 2.1. that this point will be clarified in 2.2.
- In 3.2: “PRMT activity is regulated by ubiquitylation or microRNA-mediated protein stabilization and post-translational modifications” is incorrect since microRNAs modulate the stabilization of mRNAs of PRMTs as reported in the reference 34. Thus, this is not a post-translational but a post-transcriptional modification.
- The authors should clarify how the PRMTs act on histones to epigenetically modulate gene transcription. Indeed, the only point reported is for PRMT5 in 4.3 while in 5. They claim “…The PRMT family maintains stemness and differentiation through histone modifications…”.
- In 5. The authors should explain what they mean for “…epigenetically upregulate the stemness genes….”. Sentences like this are too general.
Author Response
In the revised document, we describe the changes made in response to your helpful comments.
- However, the review is mainly focused on cancer and thus maybe in cancer stem cells rather than in stem cells. This should be better explained in the Abstract and Introduction.
Thank you for helpful comment. In abstract and introduction sections, we have been clalified that we discuss PRMT-mediated regulation in relation to mainly cancer stem cells maintenance. (page 1 lines 21–24, and page 2 lines 49–53)
page 1 lines 21–24
"Furthermore, GLI family and PRMT family have strong links with various cancers through cancer stem cell maintenance. Therefore, PRMT-mediated regulation of GLI activity would have important roles in cancer stem cell maintenance."
page 2 lines 49–53
"Interestingly, GLI family and PRMT family regulates normal and CSC function which act the origin of tissue and cancer development. Recent reports revealed PRMT1, PRMT5 and PRMT7 regulate GLI1 and GLI2 activity in normal and cancer cells. In this review, we discuss the roles of PRMT-mediated GLI regulation in relation to mainly CSC maintenance."
- In Figure 1 the repressor forms have been enclosed and in the text of 2.1. However, the explanation on how these repressor forms are formed is delayed on 2.2. The authors should anticipate in 2.1. that this point will be clarified in 2.2
Thank you for the helpful comment. The information about GLI2 and GLI3 repressor form has been clarified in the revised manuscript (page 2 lines 61–62).
page 2 lines 61–62
"Furthermore, GLI2 and GLI3 proteolytic cleavage into repressor forms (GLI2R and GLI3R) [6]."
- In 3.2: “PRMT activity is regulated by ubiquitylation or microRNA-mediated protein stabilization and post-translational modifications” is incorrect since microRNAs modulate the stabilization of mRNAs of PRMTs as reported in the reference 34. Thus, this is not a post-translational but a post-transcriptional modification.
Thank you for the helpful comment. In this sentence, "microRNA-mediated protein stabilization" has been changed to "microRNA-mediated PRMT mRNAs stabilization and translation" (page 5 line 148).
- The authors should clarify how the PRMTs act on histones to epigenetically modulate gene transcription. Indeed, the only point reported is for PRMT5 in 4.3 while in 5. They claim “…The PRMT family maintains stemness and differentiation through histone modifications…”.
Thank you for helpful comment. Modulation of gene transcription by PRMTs-mediated histone modification has been clarified in newly added section 3.4. (page 6 lines 184–204)
- In 5. The authors should explain what they mean for “…epigenetically upregulate the stemness genes….”. Sentences like this are too general.
Thank you for helpful comment. Regulation of stemness genes expression through histone arginine methylation has been explained more clearly in section 5. (page 9 lines 320–330)
Reviewer 2 Report
The review by Abe and Tanaka provides a useful insight into Gli signalling, with a focus upon protein post-translational modification as arginine methylation, and how this impacts upon gene transcription and cellular events. The authors should be commended for the production of a review that is well written and informative. Notably, the Figures produced provide a useful visual picture of the molecules involved in Gli signalling and facilitate uptake of the information covered in the text of the document. This review will be a useful source of reference for scientists working in this field and I am happy to recommend publication. There are just a few minor changes required:
Line 42 – grammatical error in the sentence.
Figure 2B – tyrosine missing an e
Figure 5 – various spelt incorrectly, decision spelt incorrectly
Author Response
Thank you for helpful comments. Grammatical error and spelling errors have been corrected in the revised manuscript.
Line 42 – grammatical error in the sentence.
In "Regulation of the GLI family is therefore strictly ocurres through multiple stages.", "is" was removed.
Figure 2B – tyrosine missing an e
Figure 5 – various spelt incorrectly, decision spelt incorrectly
Misspelled words were corrected.